# Use of Phycobiliproteins from Atacama Cyanobacteria as Food Colorants in a Dairy Beverage Prototype

**DOI:** 10.3390/foods9020244

**Published:** 2020-02-24

**Authors:** Alexandra Galetović, Francisca Seura, Valeska Gallardo, Rocío Graves, Juan Cortés, Carolina Valdivia, Javier Núñez, Claudia Tapia, Iván Neira, Sigrid Sanzana, Benito Gómez-Silva

**Affiliations:** 1Departamento Biomédico, Laboratorio de Bioquímica, Facultad de Ciencias de la Salud and Centre for Biotechnology and Bioengineering (CeBiB), Universidad de Antofagasta, Avenida Angamos N° 601, Antofagasta 1270300, Chile; juancortesgonzalez.96@gmail.com (J.C.); benito.gomez@uantof.cl (B.G.-S.); 2Departamento de Ciencias de los Alimentos y Nutrición, Universidad de Antofagasta, Avenida Angamos N° 601, Antofagasta 1270300, Chile; fran.ncisca@hotmai.com (F.S.); nutricionistavale.gd@gmail.com (V.G.); Rociograves@hotmail.com (R.G.); 3Departamento de Tecnología Médica, Universidad de Antofagasta/Avenida Angamos N° 601, Antofagasta 1270300, Chile; carovaldiviab@gmail.com (C.V.); javier.ignacio.ng@gmail.com (J.N.); claudiafran.t@gmail.com (C.T.); ivan.neira@uantof.cl (I.N.); sigrid.sanzana@uantof.cl (S.S.)

**Keywords:** cyanobacteria, phycobiliproteins, natural pigment, phycoerythrin, phycocyanin, food colorant

## Abstract

The interest of the food industry in replacing artificial dyes with natural pigments has grown recently. Cyanobacterial phycobiliproteins (PBPs), phycoerythrin (PE) and phycocyanin (PC), are colored water-soluble proteins that are used as natural pigments. Additionally, red PE and blue PC have antioxidant capabilities. We have formulated a new food prototype based on PBP-fortified skim milk. PBPs from Andean cyanobacteria were purified by ammonium sulfate precipitation, ion-exchange chromatography, and freeze-drying. The stability of PE and PC was evaluated by changes in their absorption spectra at various pH (1–14) and temperature (0–80 °C) values. Purified PBPs showed chemical stability under pH values of 5 to 8 and at temperatures between 0 and 50 °C. The antioxidant property of PBP was confirmed by ABTS (2,2′-Azino-bis (3-ethylbenzothiazoline-6-sulfonic acid) diammonium salt radical ion scavenging, and FRAP (Ferric Antioxidant Power) assays. The absence of PBP toxicity against *Caenorhabditis elegans* was confirmed up to 1 mg PBP/mL. Skim milk fortified with PE obtained a higher score after sensory tests. Thus, a functional food based on skim milk-containing cyanobacterial PBPs can be considered an innovative beverage for the food industry. PBPs were stable at an ultra-high temperature (138 °C and 4 s). PBP stability improvements by changes at its primary structure and the incorporation of freeze-dried PBPs into sachets should be considered as alternatives for their future commercialization.

## 1. Introduction

Cyanobacteria are a large, diverse and ancient group of ubiquitous Gram-negative prokaryotes that are found in terrestrial or aquatic habitats. They perform oxygenic photosynthesis and colonize freshwater, marine and brackish waters and soils and rocks from drylands. They also are found at extreme environments that are subjected to high ultraviolet radiation, high or low temperatures, desiccation, and nutrient deficiencies [1,2]. Cyanobacteria can be found as free-living unicellular or filamentous microorganisms and as photobionts in symbiotic association with fungi. Members of some filamentous genera (e.g., *Anabaena* and *Nostoc*) have heterocysts—cells specialized in atmospheric nitrogen fixation. Cyanobacteria are an important source of natural products with interest from the pharmaceutical and biotechnological industries [3,4]. Particularly, cyanobacterial pigments have attracted attention for their use in the food, textile and cosmetic industries [5,6].

Three types of cyanobacterial water-soluble phycobiliproteins (PBPs), C-allophycocyanin (APC), phycocyanin (PC), and C-phycoerythrin (PE) are organized in phycobilisomes in the photosynthetic apparatus. PE, PC and APC act as antennae pigments with absorption maxima at 562, 615, and 652 nm, respectively [7]. PBPs contain linear tetrapyrrolic chromophores (bilins) that are covalently bound to apoproteins via cysteine residues [7,8]. The ability of PBPs to act as free radical scavengers has been demonstrated to be centered on their tetrapyrrolic systems, supporting their use in the food, cosmetics and pharmaceutical industries, as well as their use as fluorochromes in biomedical research. In addition, PBPs isolated from various cyanobacterial species also have beneficial effects as anticancer, neuroprotective, anti-inflammatory, anti-allergic and hepatoprotective biomolecules [9,10,11,12,13,14,15].

Color in food has an important impact on consumers since it is one of the first characteristics we perceive from a product. Additionally, colored food is attractive, and color allows for better identification and selection among similar products. Coloration helps to relate water with food; for example, yogurt, animal or vegetal milk (e.g., soybean, coconut, or almonds) with fruits such as strawberries (reddish), blue berries (purple), and melon (green). Then, consumers would consider ingesting these types of food as beneficial to their well-being even though they may not have fruits [16].

Artificial dyes are stable at different temperatures, pH and light regimes, maintaining their coloration for long periods. Some synthetic dyes have been approved for their use in the food industry; however, some of them have been reported to be neurotoxic, mutagenic, and genotoxic (lemon-yellow tartrazine), damaging in the liver and the kidney (brilliant blue), and triggering biochemical changes and cancer in the thyroid gland (cherry-red erythrosine) [17,18,19]. As such, the demand for natural dyes in the food industry has grown in recent years due to the toxicity of artificial colorants [20]. There is great interest in finding non-harmful alternative pigments, especially blue pigments such as PC, which has the capability to scavenge hydroxyl ions in order to avoid lipoperoxidation [21]. Natural dyes can be considered renewable and sustainable bioresources with minimal environmental impact [22]. These eco-friendly, presumably mostly non-toxic, natural colorants could have applications in other industrial sectors like the cosmetics or pharmacological industries [23,24,25,26]. Other studies have shown that some natural colorants from plants have health-promoting properties in the human diet, like natural carotenoids that provide beneficial biological effects such as antioxidant and anticancer properties [27,28].

Cyanobacterial PBPs are natural pigments used as colorants in some food products; for example, aqueous extracts of non-purified blue PC from *Spirulina* have been added in ice creams, yogurts, isotonic beverages, confectionery, and jellies [29]. Particularly, bright blue PC has been selected over others less-bright natural colorants such as gardenia blue and indigo in confectionery production [30,31,32]. Red PE has been mostly used as a fluorescent probe in biomedical studies, rather than in the food industry.

*Spirulina* (*Arthrospira platensis*) has been used as a source of PBPs for additions to food products; however, this microorganism mostly produces PC. The Andes wetlands in northern Chile are a source of microbial communities that include PC and PE producing cyanobacteria. Isolated strains *Nostoc* sp. Caquena (CAQ-15), LLA-10 and *Nostoc* sp. Llayta (LLC-10) from Andes wetlands, above 3000 m of altitude, predominantly accumulate red PE, blue PC, and a purple fluorescent mixture of both, respectively. In addition, the CAQ-15 strain modulates its PBP content by complementary chromatic adaptation [33]. Here, we report the use of these isolated *Nostocaceae* strains for the purification of PBPs and their application as colorants and antioxidant molecules in the formulation of dairy functional prototypes. Additionally, we report results on the stability, antioxidant capabilities and toxicity of the purified PBPs, as well as sensory tests of the final prototypes. This work represents the first step in the use of PBPs from Andean cyanobacteria as ingredients for the food industry.

## 2. Results and Discussion

### 2.1. PBPs Stability under Different Temperature Regimes

The stability of the purified PBPs from the cyanobacterial strains LLC-10 and CAQ-15 were measured as the concentration of the remaining non-denatured PBPs after incubation at various temperature and pH regimes. Nearly 80% of the PC and PE proteins from both strains were stable after 72 h of incubation at temperatures from 10 to 21 °C. The denaturation of the PC and PE proteins increased to nearly 50% after 24 h of incubation at 26 to 53 °C (Table 1 and Figure 1). These proteins showed total denaturation after 48 h of incubation at temperatures over 55 °C (Table 1 and Figure 1). PC from both strains had a similar response to temperature; however, PE from the LLC-10 strain reached 50% denaturation at 35 °C, while PC required a lower temperature (26 °C) to reach the same level of denaturation. Additionally, PE was a thermally more stable protein (Table 1). Purified PE from the LLC-10 strain and PC from the LLA-10 strain were subjected to incubations at 138 °C for 4 s in order to evaluate changes in stability after this heat treatment. Based on changes in their visible absorption spectra, only 10% and 15% denaturation were observed at the PE and PC solutions, respectively.

The thermal stability shown by PBPs from the Atacama cyanobacterial strains was consistent with the denaturation profiles that are expected for mesophilic proteins and also for more stable proteins from thermophilic cyanobacteria [34]. PC denaturation from *Spirulina platensis* occurred after incubations over 45 °C at pH 7 [35,36,37]; the bleaching of PBPs from mesophilic cyanobacteria and algae was observed at temperatures between 60 and 65 °C [38]. Additionally, PC from *Anabaena fertilissima* PUPPCC 410.5 was unstable at 42 °C with a 50% loss after 4 days; this protein was very stable and maintained its antioxidant properties at 4 °C over 6–9 days [39,40]. Moreover, PC from *Spirulina fusiformis* was denatured at temperatures above 70 °C [41,42]. Though the PBPs from *Phormidium rubidium* A09DM were stable at temperatures from 4 to 40 °C, their corresponding absorptions at their maximal wavelengths decreased 2–4 fold at 60 to 80 °C [41]. Comparatively, PBPs from thermophilic cyanobacteria had better temperature stability than PBPs from mesophilic cyanobacteria; for example, PE from *Leptolyngbya* sp. KC 45 maintained 80% of its antioxidant activity after exposure to 60 °C for 30 min [43]. Additionally, PC from *Thermosynechococcus elongatus* TA-1 was stable between 4 and 60 °C at a pH range of 4 to 9, but PC denaturation occurred at temperatures over 75 °C [44]. Likewise, the thermostable PC from the *Synechococcus lividus* PCC 6715 strain that was isolated from a hot spring maintained 90% and 70% stability at 50 °C for 5 h and two weeks, respectively [45]. Additionally, PC from the thermophilic red algae *Cyanidioschyzon merolae* showed a midpoint denaturation at 83 °C and pH 5, with a half-life of 40 min [46].

Future work will consider the isolation of thermophilic cyanobacteria from a thermal spring in the Atacama region and the purification of their phycobiliproteins. The biochemical properties of these PBPs and relevant genetic studies would provide new unidentified protein resources for biotechnological applications.

### 2.2. PBPs Stability under Different pH Regimes

Based on their absorbance spectra, PE and PC from the LLC-10 strain were stable at pH 5 to 8. At pH 5, PE and PC showed absorbance maximum at 565 and 620 nm, respectively (Figure 2). At an acidic pH (1 to 3), a PBP precipitation was observed (Figure 2a, lower left), showing a wide non-characteristic profile of their absorption spectrum (Figure 2a, upper left). Additionally, the incubation of PBPs at an alkaline pH (9 to 14) rendered uncolored solutions with a change in their absorption spectra due to protein denaturation (Figure 2b, lower right).

Both, acidic or alkaline pH values alter the electrostatic and hydrogen bond interactions among amino acid residues in proteins; in PBP, this effect translates into structural changes in chromophores and the apoproteins [41]. The stability of Atacama PBPs from the LLC-10 strain at pH 5–8 was similar to PC from *Spirulina platensis* at pH 4–6 [35,36,37]. In addition, the addition of preservative molecules such as citric acid, sugars and calcium chloride improve PBP stability [47,48,49]. Blue PC from *Spirulina platensis* increased stability in the presence of citric acid at 35 °C over 15 days [50]. Then, non-toxic PBP stabilizers should be explored in depth to expand the use of these proteins in the food industry.

### 2.3. Antioxidants Activity

The antioxidant capabilities of purified PE and PC have been previously demonstrated. Additionally, the antioxidant power of PC has been related to its ability to sequester hydroxyl ions avoiding lipo-peroxidation [9,21,51,52,53,54].

The antioxidant activity of the purified Atacama PBPs was evaluated by two assays, ABTS and FRAP, and the results are shown in Table 2. The methanol extracts from the cyanobacterial LLA-10, CAQ-15 and LLC-10 strains showed antioxidant activity values of 195 ± 38 to 717 ± 60 µmoles Trolox equivalent (TE)/100 g fresh mass by the ABTS assay (Table 2). Purified phycocyanin (PC-LLA-10), phycocyanin (PC-CAQ-15), and phycoerythrin (PE-CAQ-15) showed antioxidant activities between 2 and 3 µmoles TE/100 mg pigment (Table 2). These results indicate that PBPs and the methanol extracts from Atacama cyanobacteria have an antioxidant power comparable to fruits such as mulberry, pineapple and passion-fruit [55]. Further support was obtained from the FRAP assay (Table 2). Consequently, cyanobacteria from the Atacama Desert are an innovative source of functional natural antioxidants that have a potential protective role against oxidative stress and biotechnological applications in the food, pharmaceutical and cosmetic industries.

### 2.4. Toxicity of Phycobiliproteins against *C. elegans*

The nematode *Caenorhabditis elegans* offers several advantages as an emerging model in environmental toxicology. It is easy and inexpensive to culture in the laboratory, it has a short life cycle that allows for short-time span experiments, and there is increasing evidence on its genetic and physiological similarity with mammals, so results related to its use have the potential to predict possible effects in higher animals [56,57]. Our work showed that PBPs that were purified from the LLC-10 strain (genus *Nostoc*) were not toxic to *C. elegans*; the nematode survival was 100% at all concentrations used; comparatively, ivermectin, a nematicidal drug, showed a 100% mortality (Figure 3), which is in agreement with the information provided by Ju et al. (2014) on other cyanobacterial pigments [58].

### 2.5. Sensory Test

Prototypes of skim milk that were fortified with the phycobiliproteins PC or PE purified from two Atacama cyanobacterial strains were the functional foods that were tested by a volunteer team by using an acceptability hedonic scale (Figure 4). The results of the sensory evaluation showed that there were no statistically significant differences between the prototypes. However, the parameters’ appearance (related to the color reached by the prototype) and texture were the best valued by judges. The appearance of the prototypes had a good acceptance (mean score 3.7) that was only surpassed by texture. The highest score at the sensory test was obtained by the skim milk that was fortified with PE (prototype N°2).

Several reports have shown a wide acceptance for food products that are supplemented with microalgal natural pigments, given the improvements in color and antioxidant properties, e.g., chlorophyll and carotenoids from *Chlorella vulgaris* and *Haematococcus pluvialis* [59,60]. In addition, microalgae have been incorporated into dairy products as a source of bioactive and coloring compounds, with good acceptability, particularly in texture and appearance [61]. In this study, texture and appearance stood out among all parameters tested. Therefore, a PE-fortified food would provide health benefits to consumers.

### 2.6. Microbiological Assays

All prototypes were subjected to a microbiological control to evaluate the potential presence of *Enterobacteriaceae*. All prototypes were free of microbial contaminant such as coliforms, *Salmonella* and *Shigella*.

## 3. Materials and Methods

### 3.1. Strains and Culture Conditions of Cyanobacteria

Diazotrophic cyanobacteria *Nostoc* sp. Llayta (LLC-10) and *Nostoc* sp. Caquena (CAQ-15), were isolated from wetlands at the XV Region in northern Chile. The strains were cultured in an Arnon liquid medium [62] without additions of combined nitrogen at 30 °C under continuous white fluorescent light irradiation (180 μE m^−2^ s^−1^) and continuous aeration with air enriched with 1% (v/v) CO_2_. Cultures were harvested by centrifugation (10 min; 6000 rpm, rotor Sorvall SS-34) at the exponential phase of growth (15 days). The liquid cultures rendered approximately 1.0 g of wet weight per 160 mL of culture.

### 3.2. Phycobiliprotein Purification

PBP purification was conducted according to the method of Ranjithak and Kaushik [63] with modifications. Briefly, cyanobacterial cell suspensions were digested overnight with lysozymes at a final concentration of 1.0 mg/mL in a 50 mM phosphate buffer of pH 7.2 at 37 °C. Next, the cell suspensions were sonicated in a water–ice bath for 2 min at 15-s intervals (Microson Ultrasonic Cell Disruptor XL, Farmingdale, NY, USA). One volume of the sonicated extract was diluted with one volume of phosphate buffer and centrifuged at 10,000 rpm for 15 min (RC 5B Plus Sorvall SS-34, Newtown, USA). The PBP-rich supernatants were recovered, and 500 μL of aliquots were used to obtain the corresponding 280–700 nm UV/VIS absorption spectrum. The selective precipitation of proteins from the remaining supernatants was conducted by adding ammonium sulfate, with continuous stirring at 4 °C to reach a 0%–35% saturation and a 35%–60% saturation to enrich precipitates with PE and PC, respectively. After 24 h, each protein precipitate was resuspended in 1.0 mL of phosphate buffer and dialyzed in 12,000 Da cut-off cellulose dialysis tubing (Sigma, Steimheim, Germany) against distilled water at 4 °C. Aliquots of the dialyzed PBP solutions were used to obtain the corresponding 280–700 nm absorption spectra. The absorption maxima 620 nm (PE), 565 nm (PC) and 650 nm (APC) were used to compute their concentration according to the following equations:[PC] (mg/mL) = ([(A615 − A730) – 0.476 (A652 − A730)] × 1/5.34)
[APC] (mg/mL) = ([(A652 − A730) – 0.208 (A615 − A730)] × 1/5.09)
[PE]] (mg/mL) = ([((A562 − A730) – 0.241 [PC] − 0.849 [APC]] × 1/9.62)

PBPs were further purified by ion exchange chromatography by using a DEAE (Diethylaminoethyl-cellulose) column (7.0 by 1.5 cm). Two or three milliliters of a PBP-dialyzed solution was loaded in the column and eluted with a 0.03–1.0 M NaCl gradient at room temperature. One-milliliter fractions were collected, and the colored fractions were pooled and scanned to obtain the UV (Ultraviolet)–Vis (Visible) absorption spectra. Concentration and purity were inferred by using the corresponding equations. The collected purified PBPs were lyophilized and stored at −20 °C until used.

### 3.3. PBPs Stability

Triplicates of each purified PBP solution were incubated in a 50 mM phosphate buffer of pH 7.2 at temperatures from 0 to 80 °C over 72 h. Protein stability was evaluated in triplicate assays at room temperature in a 50 mM sodium phosphate buffer adjusted to pH 1 to 14. Changes in PBP concentration were obtained from the corresponding UV–Vis absorption spectra every 24 h. In addition, purified PE from the LLC-10 strain and PC from the LLA-10 strain were subjected to incubations at 138 °C for 4 s; visible absorption spectra were obtained before and after the heat treatment to evaluate changes in stability.

### 3.4. PBPs Antioxidant Capabilities

The biomass from cyanobacterial strains *Nostoc* sp. Llayta (LLC-10) and *Nostoc* sp. Caquena (CAQ-15) at the exponential growth phase was recovered by centrifugation (4000 rpm, 10 min; rotor Sorvall SS-34) and washed twice with a 0.9% NaCl solution. The washed cell pellets were extracted with 4.0 mL of 70% methanol and vortexed at maximal speed for 2 min in a Genic2 multitube holder (Daigger Sci. Ind., model G560E, Bohemia, NY, USA). The methanol extracts were sonicated on a water–ice bath (Microson Ultrasonic Cell Disruptor XL, Farmingdale, NY, USA) for 3 min at 10-s intervals and clarified by centrifugation (4000 rpm, 10 min), and then the supernatants were saved. The methanol extracts were filtered through 0.2 μm SFCA (Surfactant-Free Cellulose Acetate) syringe filters (Ultra Cruz) and stored at −20 °C. The antioxidant capacity of the methanol extracts and the purified PBP was performed by ABTS and FRAP assays according to Re et al. (1999) and Benzie and Strain (1996), respectively [64,65]. Results were expressed as µmoles of Trolox equivalents per 100 g of fresh mass or milligram of pigment.

### 3.5. Toxicity Assays

Wild-type *C. elegans* var. Bristol-N2 and the *Escherichia coli* strain OP50 were obtained from the *Caenorhabditis* Genetics Center (St. Paul, MN, USA). The nematode was plate-propagated as previously described [66]. *C. elegans* var. Bristol-N2 was cultured on nematode growth agar that was seeded with either *E. coli* OP50 or *E. coli* strain B, following the procedure reported by Brenner [67]. Gravid worms were gently shaken at room temperature in 10 volumes of a fresh 1% NaClO/0.5 M NaOH solution. Carcasses and other debris were dissolved after 5–10 min, and resistant eggs (50% to 100% viable eggs) were collected and washed several times in an M-9 buffer [46]. The M-9 buffer was prepared with 1.5 g of KH_2_PO_4_, 3.0 g of Na_2_HPO_4_, 2.5 g of NaCl, 0.5 mL of 1 M MgSO_4_, and sterile distilled water to a final volume of 500 mL [68]. Eggs were deposit in agar plates for 48 h, and young worms (larval stage 4 and 5) were harvested and resuspended in the M-9 buffer.

### 3.6. Preparation and Sensory Analyses of Dairy Prototypes

Ten mL of skimmed milk were mixed with purified PBP pigments at a final concentration of 1.2 mg PC/mL and 0.3 or 1.4 mg PE/mL. The skim milk that was used in this work was a commercial liquid product prepared by Colun (Cooperativa Agrícola y Lechera de la Unión Ltd., La Unión, Chile) with a content of 3.3% of total protein, 0.05% total fat, 4.7% carbohydrates, 32 mg% sodium, 115 mg% calcium and 90 mg% phosphorus. According with the manufactures, this skim milk was processed by UHT technology (Ultra High Temperature) at 138 °C for 4 s. The acceptability of PBP-containing dairy prototypes was evaluated by sensory preferences. The study used skimmed milk as a common matrix, and three dairy prototypes were designed: Prototype N° 1 (PC from the LLC-10 strain), Prototype N° 2 (PE from the CAQ-15 strain) and Prototype N° 3 (PE from the LLC-10 strain). A five-point hedonic scale test was used to measure appearance, smell, texture and flavor. The tests involved ten university students as impartial judges who did not have previous training in sensory-type analyses. Consumer acceptability was measured with five-point hedonic scale (1—dislike extremely; 2—dislike slightly; 3—neither like nor dislike; 4—like slightly; 5—like extremely). Data were subjected to variance analysis (ANOVA) with a significance level of *p* < 0.05.

### 3.7. Microbiological Analyses

Total coliforms, *Shigella* and *Salmonella* detection in the prototypes were performed in a BBL^TM^ MacConkey II Agar (BD, Le Point Declaix, France), a selective and differential medium for the detection of coliforms and enteric pathogens. In addition, an XLD Agar (Xylose–Lysine–Deoxycholate Agar, BD, Le Point Declaix, France) was used to detect *Salmonella* and *Shigella*. Samples of each prototype were seeded on these agar plates and incubated at 37 °C for 24 h [69].

### 3.8. Statistical Analyses

Sensory test data were analyzed with an analysis of variance (ANOVA) with α = 0.05. The PBP temperature stability data were gathered from two independent experiments, each in triplicate, and a non-parametric method was used in a stability test for comparisons of means values among groups (α = 0.05). The Shapiro–Wilk normality test was done to determinate if the dataset is well-modeled by a normal distribution. Since the data did not show a normal distribution, a Kruskal–Wallis test was performed.

This research was approved by the Research Ethics Committee, CEIC, Universidad de Antofagasta (Document 044/2017).

## 4. Conclusions

PBPs purified from two Andean cyanobacteria from northern Chile showed chemical stability at a pH range from 5 to 8, and temperatures between 0 and 50 °C. The pigmented proteins from the LLC-10 strain had no toxic effects against *C. elegans*. The highest score at the sensory test was obtained by skim milk that was fortified with PE.

Colors are always attractive to children and induce consumption. The skim milk was selected to incorporate blue PC and red PE because of its low-fat content and its contribution to human health with vitamins, minerals and proteins, particularly to the overweight population.

We conclude that PBPs are natural proteins that can be used as colorants in the formulation of functional food products based on skim milk in the replacement of added artificial colorants. In addition, the native cyanobacterial PBPs have an appropriate level of antioxidant activity, and we propose their potential use as an innovative source of pigments in the pharmaceutical, cosmetic and food industries.

Finally, the UHT pasteurization (138 °C, 4 s) of Atacama cyanobacterial PBPs induced a minor denaturation of PE and PC; therefore, they can be added before the skim milk pasteurization process. However, milk usually is heated over 50 °C, and, in order to avoid the loss of added PBP coloration while maintaining bacteriological safety, we propose two alternatives for the use of cyanobacterial PBPs at high temperatures in dairy products before pasteurization. One is the application of molecular biology and genetic tools on PBP genes from mesophilic and thermophilic cyanobacteria in order to increase their stability for biotechnological processes and the safety of new functional foods. The second alternative is the use purified freeze-dried PBPs that are enclosed in sachets that can be added as powder to already pasteurized dairy food before their consumption, avoiding protein denaturation and the loss of antioxidant activity.

## Figures and Tables

**Figure 1 foods-09-00244-f001:**
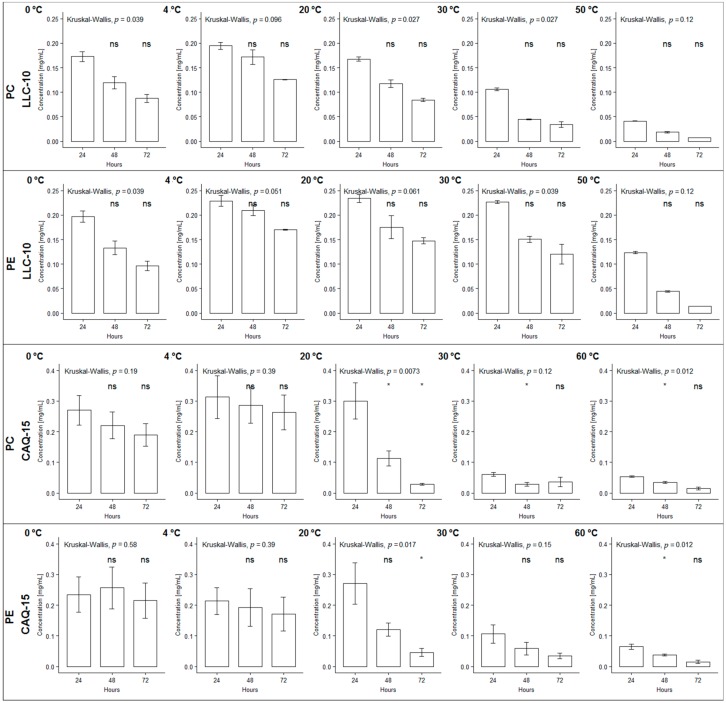
Effect of the temperature on the stability of phycocyanin (PC) and C-phycoerythrin (PE) phycobiliproteins from the cyanobacterial *Nostoc* sp. Llayta (LLC-10) and *Nostoc* sp. Caquena (CAQ-15) strains. The stability of the proteins was expressed as mg/mL of the remaining native phycobiliproteins after 24 to 72 h of incubation. (*) significant, ns: no significant with respect to the reference group at 24 h.

**Figure 2 foods-09-00244-f002:**
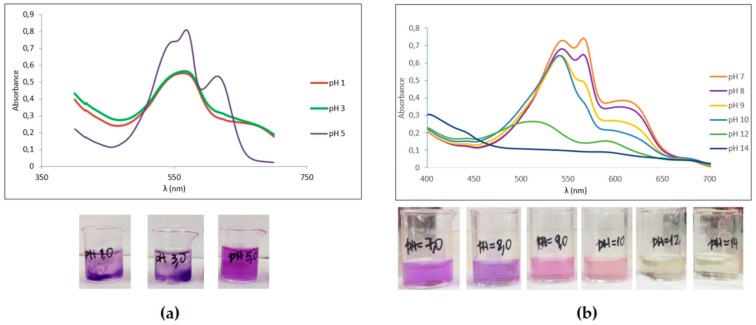
Effect of pH on phycobiliprotein stability. The stability of mixed solutions of PE plus PC from the LLC-10 strain was evaluated at an acidic (**a**), neutral and basic pH (**b**) range. Changes in absorption spectra, coloration and solubility are presented. These experiments were run in triplicate. The information provided in this figure corresponds to one complete experiment. The application of the Kolmogorov–Smirnov goodness-of-fit test showed that the inequality hypothesis was significant (*p* < 0.05) among the distribution functions for the absorption of each pH condition.

**Figure 3 foods-09-00244-f003:**
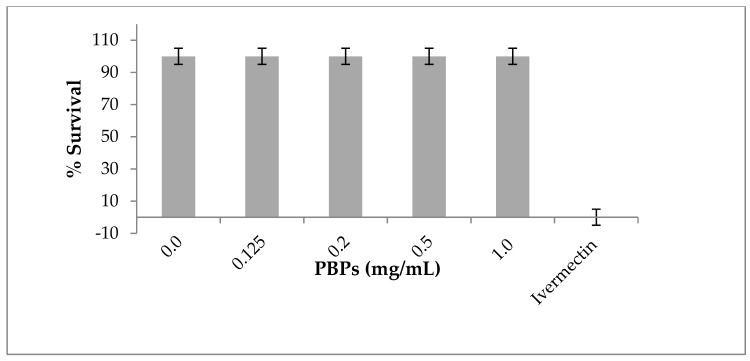
Toxicity of phycobiliproteins against the nematode *Caenorhabditis elegans*. Toxicity test of a mixed solution of PE plus PC from the LLC-10 strain was evaluated at a phycobiliprotein (PBP) concentration from 0.125 to 1.0 mg/mL. Ivermectin (0.3 mg/mL) was used as a nematicidal control drug. The nematode M-9 buffer was used as a control without phycobiliproteins. Quadruplicate tests were carried out for 24 h at 18 °C.

**Figure 4 foods-09-00244-f004:**
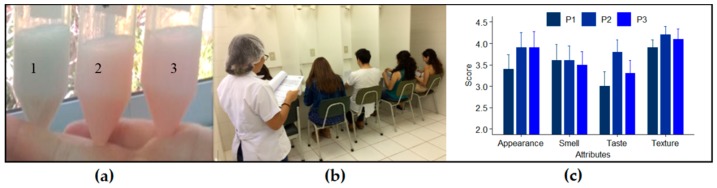
Sensory test for a PBP-containing dairy product. (**a**) Three skim milk prototypes were fortified with PE and PC: Prototype N° 1 (PC from the LLC-10 strain at 120 mg%); Prototype N° 2 (PE from the CAQ-15 strain at 13 mg%; Prototype N° 3 (PE from the LLC-10 strain at 140 mg%). (**b**) Sensory test by a volunteer team evaluating four sensory factors (appearance, smell, taste and texture) that used a consumer acceptability 5-point hedonic scale: 1—dislike extremely; 2—dislike slightly; 3—neither like nor dislike; 4—like slightly; and 5—like extremely) (**c**) Final sensory evaluation scores for four attributes of prototypes P1, P2 and P3.

**Table 1 foods-09-00244-t001:** Interpolation of temperatures to evaluate protein stability. The remaining non-denatured phycobiliproteins content was expressed as percentage (0%, 50% or 80%) of the control condition at 0 °C. Each value shown represents the interpolated temperature at 24, 48 and 72 h of incubation. The standard deviation for LLC-10 (PC), LLC-10 (PE), CAQ-15 (PC) and CAQ-15 (PE) were 9.3 to 13.4, 13.6 to 14.2, 15.1 to 16.4, and 15.72 to 18.6, respectively.

Strain PBP		Percentage of Remaining Phycobiliproteins at Different Interpolated Temperatures (°C)								
		0%			50%			80%		
		24 h	48 h	72 h	24 h	48 h	72 h	24 h	48 h	72 h
LLC-10	PC	84.7	54.3	52.8	42.3	27.1	26.4	16.9	10.9	10.6
	PE	106.9	76.9	70.4	53.4	38.5	35.2	21.4	15.4	14.1
CAQ-15	PC	68.1	57.8	49.5	34.0	28.9	24.7	13.6	11.6	9.9
	PE	82.7	62.5	52.5	41.4	31.3	26.2	16.5	12.5	10.5

**Table 2 foods-09-00244-t002:** Antioxidant activity of purified phycobiliproteins and methanol extracts from the Atacama native cyanobacterial strains CAQ-15, LLC-10 and LLA-10. The antioxidant capabilities were evaluated by the ABTS and FRAP assays and expressed as TEAC (Trolox equivalent antioxidant capacity). TE: Trolox equivalents; PE: Phycoerythrin; and PC: Phycocyanin. The assays were conducted in triplicate, and the results are shown as the mean values with the corresponding standard deviation.

Sample	ABTS	FRAP
Methanol extract	µmoles TE/100 g fresh biomass	
CAQ-15	717 ± 61	50.23 ± 1.64
LLC-10	641 ± 95	12.63 ± 0.73
LLA-10	195 ± 38	13.14 ± 1.52
Phycobiliprotein	µmoles TE/100 mg phycobiliprotein	
PE-CAQ-15	198 ± 45	0.92 ± 0.15
PC-CAQ-15	312 ± 56	1.55 ± 0.10
PC-LLA-10	205 ± 41	2.50 ± 0.15

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
