# Peer review of "Use of Phycobiliproteins from Atacama Cyanobacteria as Food Colorants in a Dairy Beverage Prototype"

_foods, 2020, doi:10.3390/foods9020244_

Round 1
Reviewer 1 Report
The authors have revised sufficiently .
Reviewer 2 Report
The authors addressed all the reviewers' concerns about the manuscript.
Reviewer 3 Report
The authors have revised.
This manuscript is a resubmission of an earlier submission. The following is a list of the peer review reports and author responses from that submission.
Round 1
Reviewer 1 Report
Personally, I think that the general idea is good, particularly in the perspective of replacing synthetic colorants with those from natural origin. Therefore, I suggest the Editor to consider this paper for publication, though some revision are needed. Indeed, I think that the work can be improved. In my opinion, the authors perfromed a great number of experiments without a clear design or goal. Below are my considerations on this research work:
first of all I would spend more words on the significance of the work. It is not clear to me if the advantage of using colorants from cyanobacteria is that they are natural and not synthetic or if thir addition to foods may have some other functionalities. I think authors should better explain the main goal of their study. A comprehensive description of these specific microrganism is missing in the introduction. I suggest the authors to add a brief and exhaustive description of the microrganism used in the study. The authors stated that PBPs from cyanobacteria have anti-oxidant power on the basis of the ABTS test. I strongly reccomend the authors to test the anti-oxidant capacity also by means of other assays (i.g. DPPH, FRAP and ORAC tests) as the use of only one of these in vitro assays is not really indicative of the anti-oxidant potential. For what concern the prototype of the dietary product colored with PBPs, I could not find the reason why the authors chase milk as the matrix. Why should dietary industries color milk? What is the advantage of having a colored milk instead of white milk? The authors should clearly state the reason why they decided to develop a colored food product based on milk.
Reviewer 2 Report
The manuscript "Use of phycobiliproteins from Atacama cyanobacteria as a food colorant" by Galetović and others, describes some properties of phycobiliproteins extracted from cyanobacteria with potential to be used as food pigments. The different phycobiliproteins fractions had different stability under various temperatures and pH values. Furthermore, the extracts were used in a dairy beverage model. While the manuscript provides insightful information about novel natural pigments, some aspects need to be improved to be suitable for publication in Foods.
Please re-write the title in a more informative way to allow the readers to understand better the context. For example: "Use of phycobiliproteins from Andean cyanobacteria as food colorants in a dairy beverage model" Summarize the introductory part of the abstract and include a sentence at the end providing expectations of future studies. Avoid the use of very long sentences throughout the manuscript. In the introduction, please indicate if there are any previous studies available about the use of phycobiliproteins as food pigments. If there are not, highlight the novelty of this manuscript. In Fig. 1 caption, please clarify the statistical analysis. It is not clear what statistical analysis was performed. Additionally, indicate what “ns” means, not significant compared to what? Please provide statistical analysis for Table 1, Table 2, and Figure 2 results. Indicate why PE-LLA was not analyzed for antioxidant activity and sensory evaluation. The sensory data is not available, please provide a table or figure to indicate the results of the sensory evaluation. Please indicate in the methodology, what criteria were used to establish the extracts’ concentration in the dairy beverage. In the statistical analysis section of the methodology, clarify why a non-parametric statistical analysis was used. Correct typos throughout the manuscript.Reviewer 3 Report
The paper on the Use of phycobiliproteins from Atacama cyanobacteria
3 as a food colorant is an interesting one however, the authors have rushed into its publication and need to take into account of the following before publication.
First the structure of the publication. Materials and methods usually goes before results and discussion.
Introduction needs more references.
Extensive English correction is required.
The authors mention a functional food product a skim milk but they do not say anything about it, is it skimmed milk, is it skimmed milk powder. They say in the figure
Prototypes of skim milk fortified ( Prototype N° 2: PE from strain CAQ-15; 13 mg% and 179 Prototype N° 3: PE from strain LLC-10; 140 mg%. What are these concentrations? What is the protein and fat percentage of these products?
Before carrying out a sensory analysis the product needs to be safe.
I have not seen any microbiological analysis tests (total microbial count, coliforms, salmonella) before doing what the authors mention-Future studies must be designed to improve the stability of PBPs at high temperatures in order to guarantee the safety of their use in new food products.
Hence, the manuscript does not comply with the journal's requirements and should be rejected.